

# Mitochondrial and *Wolbachia* phylogenetics of the introduced Jorō spider, *Trichonephila clavata* (Araneae: Araneidae) in North America

James E. Russell[1], Nicholas Mizera[1], Christopher G. Brown[1], Angela Chuang[2], David R. Coyle[2] and David R. Nelsen[3]

[1] Department of Biological Sciences, Georgia Gwinnett College, Lawrenceville, GA, United States of America
[2] Department of Forestry and Environmental Conservation, Clemson University, Clemson, SC, United States of America
[3] Department of Biology and Allied Health, Southern Adventist University, Collegedale, TN, United States of America

## ABSTRACT

The introduction of *Trichonephila clavata* (L. Koch, 1878) (Araneae: Araneidae: subfamily Nephilinae) in the United States was first recorded in Georgia in 2014. Since its introduction, *T. clavata* has become a prominent feature of the arthropod fauna in several southeastern US states. Many questions regarding the introduction event(s) remain unanswered; for instance, was the introduction a single discrete event followed by rapid spread, or were there multiple introductions? The mitochondrial cytochrome c oxidase subunit one gene region (COI), which was used to characterize the initial *T. clavata* observation in the US, has also been used to characterize within- and between-population genetic variation. One confounding factor for COI as a population genetic molecular marker, though, is the presence of cytoplasmic agents of selection such as intracellular bacteria in the genus *Wolbachia*. Given that *Wolbachia* infections have been detected in potential source populations of *T. clavata*, the present study sought to characterize mitochondrial genetic diversity and the status of *Wolbachia* infection in the North American population(s) closest to the originally proposed introduction site in Georgia. DNA sequencing revealed no mitochondrial genetic variation in the *T. clavata* population sampled in North America, and an exact sequence match to the previously reported *T. clavata* in Georgia and a sequence sample from Yunnan, China. *Wolbachia* was detected in the North American samples. However, phylogenetic analysis on a concatenated multi-locus type sequence suggested two distinct *Wolbachia* clades, one represented by samples collected in Georgia and another represented by a single sample collected in South Carolina. Sequence analyses of the multi-locus gene regions suggested that the Georgia *T. clavata* may be infected with two strains of *Wolbachia* (super-infection), and the South Carolina sample represented a separate single infection. The study's results emphasize the need for further research, including expanded sampling in the introduced and potential source population regions, as well as a more detailed molecular characterization of the populations.

Corresponding author
James E. Russell, jrussell@ggc.edu

# INTRODUCTION

*Trichonephila clavata* (L. Koch, 1878) (Araneae: Araneidae: subfamily Nephilinae) is an orbweaver spider native to eastern Asia, including Japan, Korea, and China (*Hoebeke, Huffmaster & Freeman, 2015*). The phylogenetic placement of the genus *Trichonephila* and other nephiline spiders relative to other Orbipurae orbweavers is a current topic of debate, with alternative placements in either their own family, Nephilidae (see *Kuntner, 2006*; *Kuntner et al., 2019*; *Kuntner et al., 2023* and arguments within) or as a subfamily, Nephilinae, synonymized within the extremely large family of Araneidae (see *Dimitrov & Hormiga, 2009*; *Dimitrov et al., 2017*; *Scharff et al., 2020*; *Kallal et al., 2020*; *Hormiga et al., 2023*).

In late 2014, the first *T. clavata* were collected in Georgia, US (*Hoebeke, Huffmaster & Freeman, 2015*). This species, a congener of the native *T. clavipes* (Linnaeus, 1767; the golden silk orb weaver), quickly spread across the southern US in the years after its introduction, with one main population spanning Georgia, Tennessee, South Carolina, and North Carolina (*Chuang et al., 2023*), as well as recent observations which may represent new populations in Maryland, Pennsylvania, Virginia, and Massachusetts (https://john-deitsch.shinyapps.io/joroshiny/). Surveys on local orbweaver communities suggest there are fewer native orbweavers where *T. clavata* has been established the longest (*Nelsen et al., 2023*). As such, *T. clavata* requires further close observation and research in the North American habitats to determine its impacts. With overall research still in the early stages and a dearth of population genetic information despite the potential distribution of *T. clavata* across much of North America (*Davis & Frick, 2022*; *Nelsen et al., 2023*), questions remain about the introduction of this highly visible invasive species. Was *T. clavata* introduced to North America through a single introduction event? Could multiple introduction events be partially responsible for the species' already large range in North America?

Population genetic tools can provide a better understanding of the past and future of *T. clavata* in North America. For example, when founded by a few closely related individuals, little or no genetic diversity is expected in a population, which restricts that population's adaptive potential (*Booy et al., 2000*)—a condition that could be particularly pertinent for an introduced species in a novel environment. In 2014, the mitochondrial gene cytochrome c oxidase subunit I (COI) from these new North American samples was compared to the recorded literature and confirmed the species identification of *T. clavata* (*Hoebeke, Huffmaster & Freeman, 2015*). COI is useful as a "barcoding" gene, serving as a common identifier of species, and often has several distinct alleles in a population (*Hebert et al., 2003*), making this gene region particularly useful for both within- and between-population studies. Though COI has been used in many population-level studies of animals (*Tavares et al., 2011*; *Karthika et al., 2017*; *Beebe, 2018*), the use of any genetic marker is limited by the extent of prior study and the authenticity of previous sequences, often limiting its usefulness outside of already extensively-studied species (*Dawnay et al., 2007*). Additionally, mitochondrial population genetics can be confounded by many

factors (*Galtier et al., 2009*), including highly oxidative environments, complex mutation processes, and other cytoplasmic agents of selection.

Bacteria in the genus *Wolbachia* (Alphaproteobacteria, Rickettsiales) can function as agents of selection. *Wolbachia* are cytoplasmic bacterial symbionts well represented across arthropod taxa, with one estimate suggesting that up to 66% of all insects may be infected (*Hilgenboecker et al., 2008*). In arthropods, *Wolbachia* exhibits several forms of genetic drive that selectively favor infected hosts through reproductive manipulations that either create reproductive barriers between infected and uninfected individuals or distort sex ratios in a manner that promotes the spread of *Wolbachia* within populations (*Stouthamer, Breeuwer & Hurst, 1999*). Given that cytoplasmic genetic elements are typically co-inherited, the result of *Wolbachia* infection in many arthropod populations is indirect selection on mtDNA (*Turelli, Hoffmann & McKechnie, 1992*; *Jiggins, 2003*; *Narita et al., 2006*). This can result in selective sweeps of *Wolbachia* and co-inherited mtDNA that reduce mitochondrial genetic diversity, as has been characterized in *Drosophila simulans* (*Turelli, Hoffmann & McKechnie, 1992*), *Acraea encedon* (*Jiggins, 2003*), and many other insect species (*Cariou, Duret & Charlat, 2017*). However, *Baldo et al. (2008)* found that strict cytoplasmic co-inheritance among *Agelenopsis* spiders was disrupted by apparent horizontal transfer among and within species, with some mitotypes associating with divergent *Wolbachia* strains, and individual *Wolbachia* strains associating with divergent mitotypes.

*Wolbachia* have been detected in Chinese and Korean populations of *T. clavata* (*Wang et al., 2010*; *Yang et al., 2021*; *Oh et al., 2000*). Using strain-specific primers for the *Wolbachia wsp* gene region, *Wang et al. (2010)* discovered a double-infected *T. clavata* population in Wuhan, Hubei Province, China. Furthermore, *Yang et al. (2021)* characterized a single infection of *T. clavata* collected in Mangshan, Guangdong Province, China, using multi-locus sequence typing of gene regions (*Baldo et al., 2006*). These studies were part of large-scale *Wolbachia* prevalence investigations among spider species and did not sample *T. clavata* populations to any substantial degree (fewer than ten samples in each study), nor did they examine the potential impact of *Wolbachia* on host mitochondrial population genetics.

Our study took a phylogenetic approach, using mitochondrial and *Wolbachia* molecular markers, to investigate the founding population of *T. clavata* in the US. We addressed three questions: (1) What is the mitochondrial genetic diversity of the southeastern *T. clavata* population in North America? (2) Is the introduced population infected with *Wolbachia*? and (3) If the introduced population is infected with *Wolbachia*, to what extent is mitochondrial and *Wolbachia* genetic diversity linked? Given the recency of the introduction event, we hypothesized low levels of genetic diversity among *T. clavata* and associated low levels of *Wolbachia* diversity, assuming coinheritance between mitochondria and *Wolbachia*.

## MATERIALS & METHODS

### Sample collection

Mature female *T. clavata* were collected between August and December 2022 from various natural and managed locations (Table 1; Fig. 1) and mature female *T. clavipes* were collected

**Table 1** *Trichonephila clavata* **sample names and locations.** Data shown below depicts the abbreviations of samples used in the study, as well as city of origin, state of origin, and the specific sample designation of samples tested for *Wolbachia* detection if applicable.

| Abbreviation | City | State | Replicate # | Wolb. sample |
|---|---|---|---|---|
| AGA | Athens | GA | 3 | AGA1 |
| APGA | Buford | GA | 3 | APGA1 |
| APGB | Buford | GA | 1 | APGB |
| AUGA | Auburn | GA | 3 | AUGA3 |
| BGA | Braselton | GA | 4 | BGA2 |
| BUGA | Buford | GA | 3 | BUGA2 |
| CCGB | Jefferson | GA | 1 | CCGB |
| CLSC | Clemson | SC | 3 | CLSC1 |
| CPGB | Athens | GA | 1 | CPGB |
| DDGA | Decatur | GA | 5 | DDGA1 |
| RGA | Richland | GA | 3 | RGA3 |
| SGA | Suwannee | GA | 3 | SGA3 |
| TGA | Talmo | GA | 4 | TGA4 |
| WGA | Watkinsville | GA | 4 | WGA4 |
| WIGA | Winder | GA | 3 | WIGA1 |
| CTN | Collegedale | TN | 1 | N/A |
| CGA | Calhoun | GA | 3 | N/A |
| WBGA | Woodstock | GA | 4 | N/A |
| JGA | Jefferson | GA | 3 | JGA1 |
| GGA | Gainesville | GA | 3 | GGA3 |
| GGGB | Lawrenceville | GA | 1 | GGGB |
| FMGA | Lawrenceville | GA | 3 | FMGA1 |
| FMGB | Lawrenceville | GA | 1 | FMGB |
| HGA | Hoschton | GA | 3 | HGA2 |
| LGA | Loganville | GA | 3 | LGA2 |
| LMGB | Auburn | GA | 1 | LMGB |
| MGA | Morrow | GA | 5 | N/A |
| MGB | Morrow | GA | 4 | MGB1,2,4 |
| HEGA | Helen | GA | 3 | HEGA1 |
| HPGA | Dacula | GA | 4 | HPGA4 |
| SMGA | Hoschton | GA | 4 | N/A |
| SPGB | Hoschton | GA | 1 | SPGB |
| TMGB | Lawrenceville | GA | 1 | TMGB |
| FBGA | Flowery Branch | GA | 4 | N/A |
| MAGA | Marietta | GA | 2 | N/A |
| FNC | Fayetteville | NC | 2 | N/A |

in North Carolina before *T. clavata* invaded the area. Spiders were collected individually and stored separately by location and frozen at −20 °C. The collection of spider samples was approved by a Georgia Gwinnett College STEC 4500 research agreement.

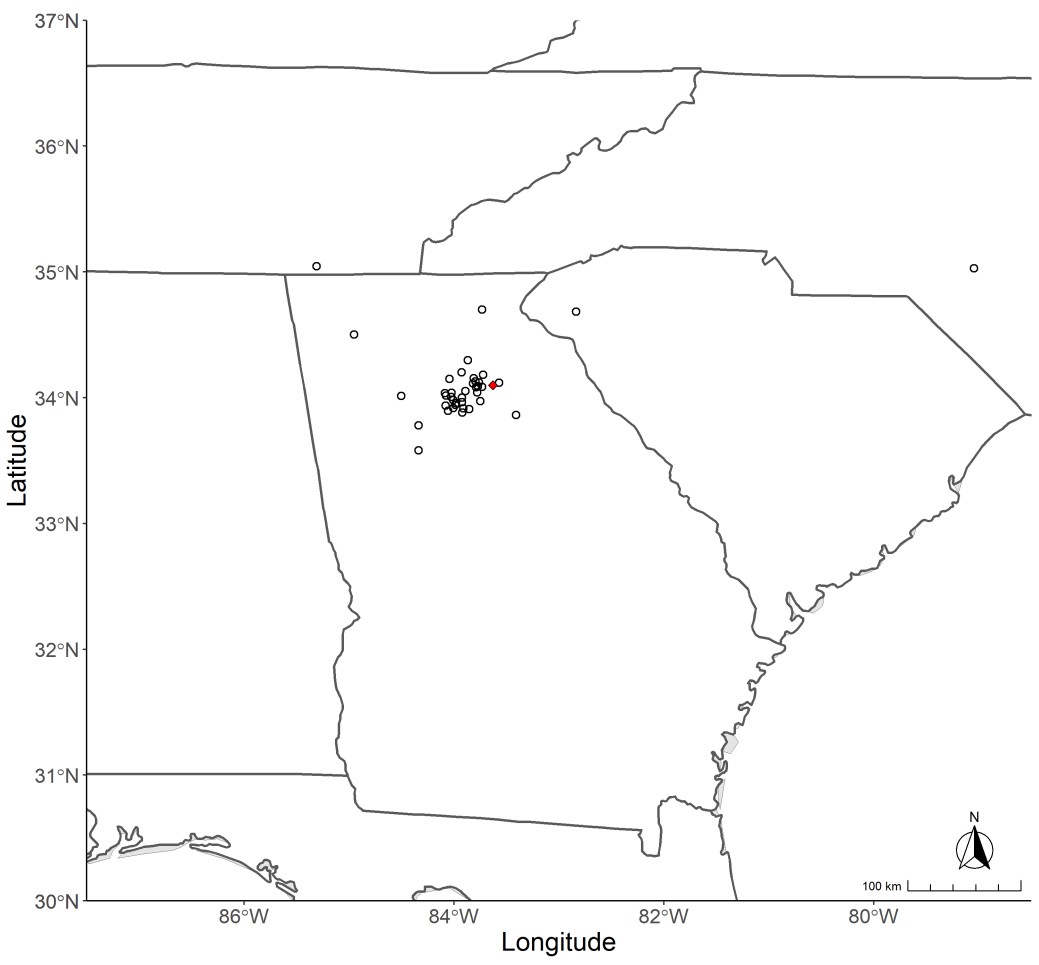

**Figure 1 Map of collection sites for *Trichonephila* species.** The states from which *Trichonephila* spp. were sampled is shown above. State outlines containing polygons indicating collection sites are clockwise from top left: Tennessee, North Carolina, South Carolina, and Georgia. Red dot indicates the approximate location where *T. clavata* were initially found in 2014.

## Molecular methods

Single-leg tissue samples were used for DNA extraction using 5% Chelex (Bio-Rad Laboratories, Hercules, CA, US) according to modified methods described in *Walsh, Metzger & Higuchi (1991)*. PCR reactions contained one microliter of template DNA, 12.5 microliters of Promega GoTaq® master mix, one microliter of each 10 µM primer, and 9.5 µl of molecular-grade water. A portion of the cytochrome oxidase subunit one (COI) gene region was amplified using primers LCO (5′-GGTCAACAAATCATAAAGAT ATTGG-3′) and HCO (5′-TAAACTTCAGGGTGACCAAAAAATCA-3′) (*Folmer et al., 1994*). Thermocycle conditions consisted of an initial denaturation at 95 °C for four minutes, followed by 34 cycles of 94 °C for forty-five seconds, 55 °C for thirty seconds, 72 °C for one minute and thirty seconds, followed by a final extension at 72 °C for ten minutes.

**Table 2  Sample names, sample locations, and source material for all (A) COI and (B) *Wolbachia* MLST sequences acquired as supplements to phylogenetic reconstructions using *T. clavata* from samples collected in Georgia and South Carolina (NA = information not available).**

| Sample name | Location | Source |
| --- | --- | --- |
| **(A) Cytochrome oxidase subunit one (COI)** | | |
| Georgia 1 | Georgia, USA | *Hoebeke, Huffmaster & Freeman (2015)* |
| Georgia 2 | Georgia, USA | *Hoebeke, Huffmaster & Freeman (2015)* |
| Guangdong 1 | Guangdong, China | *Yang et al. (2021)* |
| Yunnan 1 | Yunnan, China | *Hoebeke, Huffmaster & Freeman (2015)* |
| Yunnan 2 | Yunnan, China | *Hoebeke, Huffmaster & Freeman (2015)* |
| Zhejiang 1 | Zhejiang, China | *Hoebeke, Huffmaster & Freeman (2015)* |
| Zhejiang 2 | Zhejiang, China | *Hoebeke, Huffmaster & Freeman (2015)* |
| South Korea 1 | South Korea | *Hoebeke, Huffmaster & Freeman (2015)* |
| Taiwan 1 | Taiwan | *Hoebeke, Huffmaster & Freeman (2015)* |
| **(B) *Wolbachia* Multiple Locus Sequence Types (MLST)** | | |
| *Araneus ventricosus* | Guangdong, China | *Yang et al. (2021)* |
| *Mesida yini* | Guangdong, China | *Yang et al. (2021)* |
| *Pardosa mionebulosa* | Guangdong, China | *Yang et al. (2021)* |
| *Trichonephila clavata* | Guangdong, China | *Yang et al. (2021)* |
| *Agelenopsis aperta* | Oklahoma, USA | MLST database |
| *Acraea encedon* | NA | MLST database |
| *Aedes albopictus* | Koh Samui, Thailand | MLST database |
| *Armadillidium vulgare* | NA | MLST database |
| *Brugia malayi* | NA | MLST database |
| *Drosophila melanogaster* | NA | MLST database |
| *Nasonia giraulti* | NA | MLST database |
| *Rhagoletis cerasi* | Czech Republic | MLST database |

*Wolbachia* multi-locus sequence typing (MLST) primers for *coxA*, *fbpA*, *gatB*, and *hcpA* identified by *Baldo et al. (2006)* were used to amplify the respective gene regions using the previously established protocols. *Wolbachia* A and B supergroup-specific *w*sp primers (*Zhou, Rousset & O'Neill, 1998*) were used to identify the spider's superinfection status. All PCR products were analyzed with 2% agarose gel electrophoresis, and successful amplicons were prepared for sequence analysis with ExoSAP-IT® (Applied Biosystems, Waltham, MA, US). Of the 94 *T. clavata* samples successfully amplified for COI, a subset of 31 were used to amplify *Wolbachia*-specific gene regions, including the MLST gene regions.

Additional COI and MLST sequences were acquired from accession numbers provided from *Hoebeke, Huffmaster & Freeman (2015)*, *Yang et al. (2021)*, and the *Wolbachia* MLST database (http://www.pubmlst.org/wolbachia/) (Tables 2A–2B).

### Sequence analysis

All successfully cleaned amplicons were sent to Eurofins Genomics (Louisville, KY, US) for Sanger sequencing. Received sequence reads were aligned in MUSCLE using default parameters. Chromatograms (https://technelysium.com.au/wp/chromas/) were used to inspect sequence reads by eye and verify polymorphisms. Cytochrome oxidase subunit one (COI), all MLST *Wolbachia* loci, and concatenation of MLST sequences

were phylogenetically analyzed in BEAST (*Suchard et al., 2018*) using Bayesian inference methods and in MEGA 11 (*Tamura, Stecher & Kumar, 2021*) using maximum likelihood methods with 1000 bootstrap replications (see Supplementary Materials). Appropriate evolutionary models were determined using the model test function in MEGA 11 and the Akaike information criterion scores. BEAST parameters for the COI analysis included the Tamura-Nei substitution model (*Tamura & Nei, 1993*), a chain length of 10,000,000 generations with log parameters saved every 1,000 generations and a burn-in of 3,000 samples. All *T. clavata* sequences were recovered in a monophyletic clade with a posterior probability of 1.0. Parameters for the concatenated *Wolbachia* MLST analysis were identical to the COI analysis with the exception of a Hasegawa-Kishino-Yano substitution model (*Hasegawa, Kishino & Yano, 1985*). BEAST generated trees were visualized in FigTree v1.4.4 (http://tree.bio.ed.ac.uk/software/figtree/).

## RESULTS

### COI

Spider samples from one collection location, Tennessee, failed to successfully amplify and were excluded from subsequent analyses. The North Carolina collection location produced two *T. clavipes* samples, the only congener to *T. clavata* in North America. Three of the 94 *T. clavata* samples sent for sequencing were not readable and were excluded from analysis, resulting in a final $n = 91$. No polymorphisms were observed among the 91 *T. clavata* COI sequence samples from Georgia and South Carolina. The haplotype was identical across the 620 sites analyzed to samples collected in Georgia (*Hoebeke, Huffmaster & Freeman, 2015*) and to Yunnan 1 (HQ441928.1). The COI tree (Fig. 2) resolved two main *T. clavata* clades with a posterior probability of 1.0, one of which included all US samples, Yunnan 1, Guangdong, and Zhejiang 1, and a sister clade that included South Korea as an outgroup to Yunnan 2, Zhejiang 2, and Taiwan. Maximum likelihood analysis returned a similar topology with a *T. clavata* polytomy that included support for a US + Yunnan 1 clade and a clade that included South Korea, Yunnan 2, Zhejiang 2, and Taiwan. Guangdong and Zhejiang 1 were unassigned to either clade in the polytomy (see Material S1). Given the limited sample size, taxonomic breadth, and estimated recent evolutionary time frame for the taxa used in the COI analyses, low levels of support for some of these clades are not unexpected.

Trichonephila clavipes samples FNC1 and FNC2 served as an outgroup for the *T. clavata* clade and were used to root the tree. Due to the genetic monomorphism of sampled *T. clavata*, only those samples that were also used in the *Wolbachia* analysis were reported for COI phylogenetic analysis (see Material S2 table for accession numbers).

### *Wolbachia*

*Wolbachia* supergroup A and B-specific *wsp* primers (*Zhou, Rousset & O'Neill, 1998*) were used to detect superinfection among collected samples. Although the amplification success of the two primer regions was limited, both primers successfully amplified some of the Georgia samples, suggesting superinfection of A and B supergroup *Wolbachia* in those samples (8/8 or 100% A-specific amplification, 22/31 or 71% B-specific amplification, with

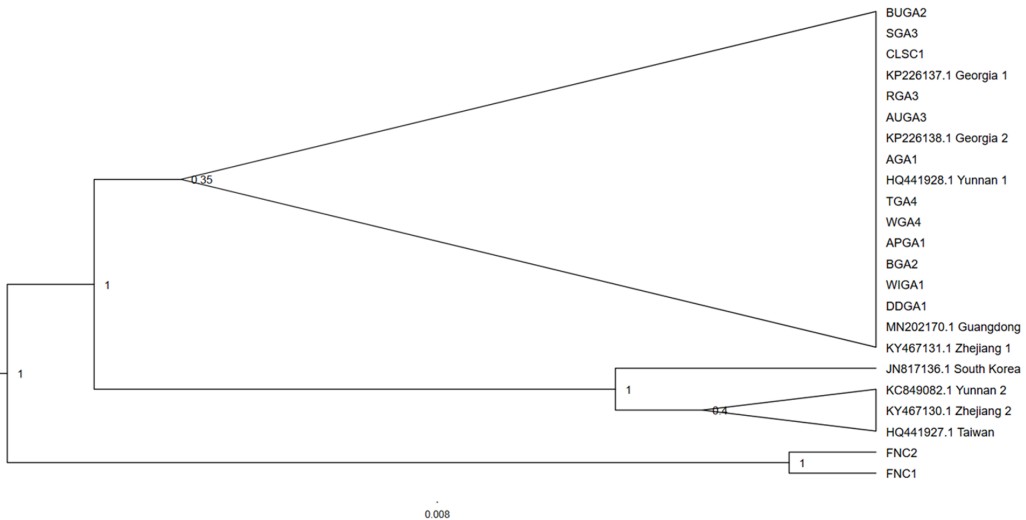

**Figure 2** **Cladogram of the cytochrome oxidase subunit one (COI) gene region of *Trichonephila clavata*.** Bayesian inference phylogenetic tree for the cytochrome oxidase subunit one (COI) gene region of *Trichonephila clavata* obtained in Georgia (GA) and South Carolina (SC). Reference samples obtained in Genbank are included with accession numbers. *Trichonephila clavipes* (FNC) was used to root the tree. Posterior probabilities are indicated at nodes. *T. clavata* samples, within the collapsed clade with posterior probability of 0.35, represent a subset of 91 samples sequenced that were identical.

all A-specific amplifications also amplifying the B-specific gene region). One sample used in phylogenetic analysis, CLSC1, failed to amplify either *wsp* gene region.

Due to limited success at amplifying all MLST loci, a subset of *T. clavata* samples and MLST loci (*wsp* and *ftsZ* sequences were omitted) were used for analysis. *Wolbachia* supergroup A and B specific primers (*Zhou, Rousset & O'Neill, 1998*) successfully amplified a further subset of *T. clavata* samples but were not used for sequencing. Concatenation of MLST loci *coxA*, *fbpA*, *gatB*, and *hcpA* among 12 collected *T. clavata* samples was used for phylogenetic analysis (see Material S2 table for accession numbers). The gene regions chosen for concatenation were selected based on visual inspection of the chromatograms.

All observed polymorphisms in *T. clavata* were bimodal. Upon visual inspection of polymorphisms, double peaks corresponding to the alternative allele were found for all loci, a result that could be indicative of double infection. CLSC, the one sample from South Carolina, was unique among the *T. clavata* samples with no observed double peaks at polymorphic sites and an identical genotype to *T. clavata* from China (*Yang et al., 2021*) for three of the four MLST loci (*coxA*, *gatB*, *hcpA*) used in this study (Fig. 3). The nearest match allele profiles, based on the *Wolbachia* MLST database, for CLSC were unique among the sampled *T. clavata* for all loci except hcpA. All other samples shared nearest match allele profiles with other sampled *T. clavata* for three out of the four MLST loci.

Phylogenetic analysis of the concatenated *Wolbachia* MLST loci, with *Brugia mayali* serving as an outgroup, returned two deeply branched clades, which can be attributed to *Wolbachia* supergroups A and B (Fig. 4). All sampled *T. clavata* are recovered in the *Wolbachia* A supergroup. The monophyletic Georgia *T. clavata Wolbachia* clade with

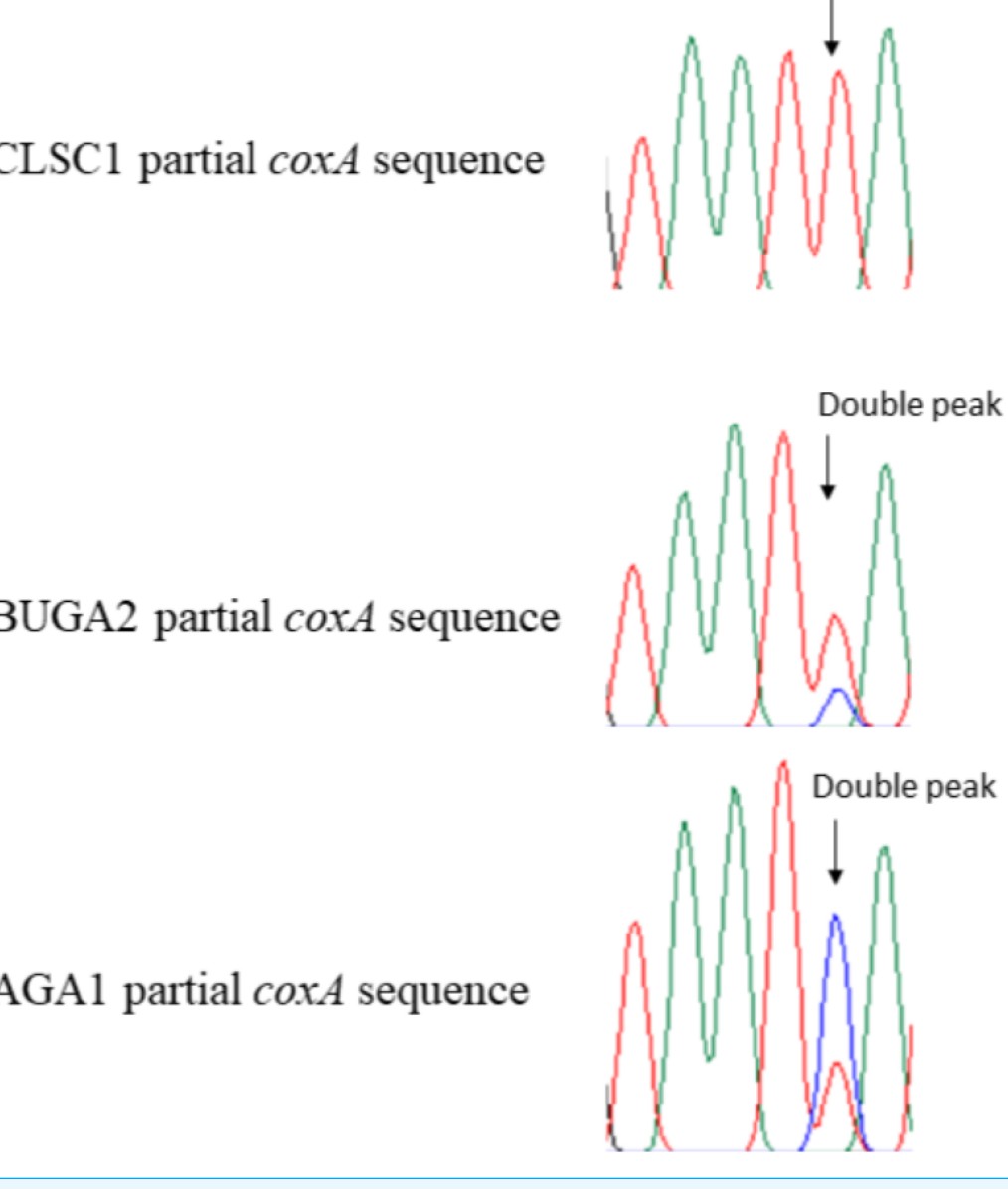

**Figure 3** **Partial chromatogram results from the *Wolbachia coxA* gene region of *Trichonephila clavata*.** Examples of distinguishing features for South Carolina (CLSC1) and Georgia (BUGA2 and AGA1) *Wolbachia* MLST sequence results are shown in partial *coxA* chromatograms above. Polymorphic *Wolbachia* MLST sequence sites were represented by single peaks in the South Carolina sample, and double peaks in Georgia samples.

*Wolbachia* from the spider species *Agelenopsis aperta* as sister, was phylogenetically distinct from the clade containing the South Carolina sample, CLSC (posterior probability 1.0). The South Carolina MLST profile was sister to *T. clavata* from *Yang et al. (2021)* and shared a clade with *Wolbachia* MLST sequences derived from the parasitic wasp *Nasonia giraulti*. Maximum likelihood analysis returned generally similar results regarding the disjunct

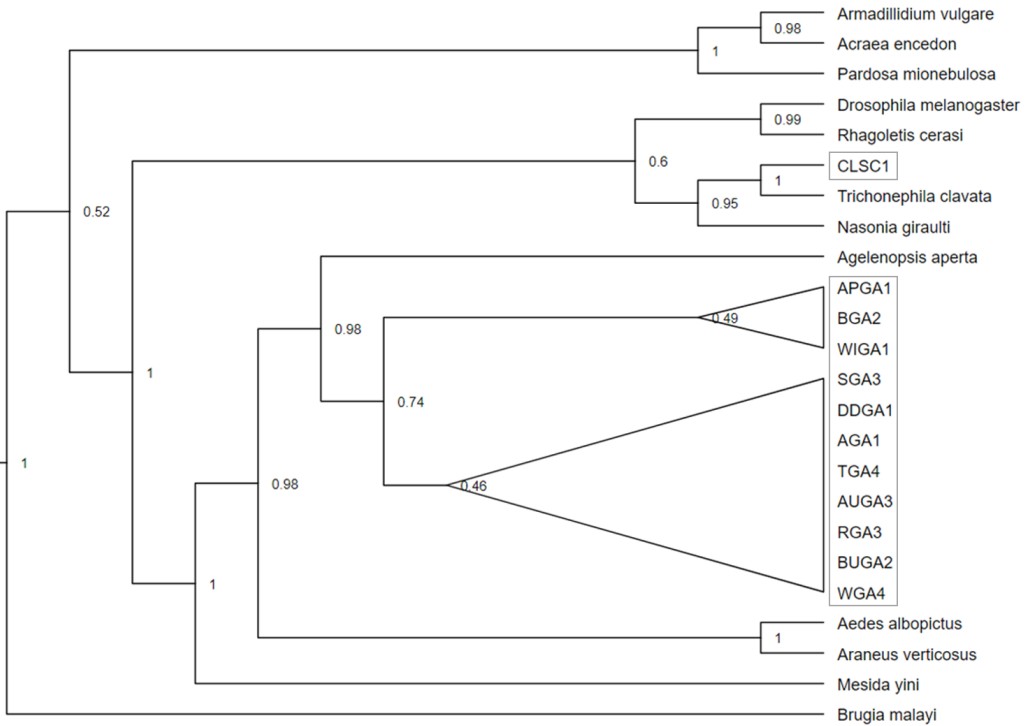

**Figure 4  Cladogram of a concatenation of *Trichonephila clavata Wolbachia* MLST gene regions.**
Bayesian inference phylogenetic tree for concatenated *Wolbachia* gene regions *coxA*, *gatG*, *fbpA*, and *hcpA*
of *Trichonephila clavata* obtained in Georgia (GA) and South Carolina (SC), indicated with grey outline.
Reference samples indicated by genus species were obtained from the Wolbachia MLST database. *Brugia
mayali* was used as an outgroup to root the tree.

topology of the US *T. clavata Wolbachia* with a polytomy that included a Georgia clade
(75% bootstrap support) and a poorly supported clade (59%) that included South Carolina
and *T. clavata* from *Yang et al. (2021)* (Material S3).

## DISCUSSION

Phylogenetic analysis of *T. clavata* from the US (focused on Georgia and South Carolina)
reveals contrasting patterns of cytoplasmic diversity for the mitochondrial (COI) and
*Wolbachia* MLST loci (*coxA*, *fbpA*, *gatB*, and *hcpA*) (Figs. 2 and 4). Like the previous limited
analysis by *Hoebeke, Huffmaster & Freeman (2015)*, which was primarily used to confirm
the species status of the recently discovered spider in the US, no mitochondrial genetic
diversity was observed in the larger sample size of the present study. These findings suggest
a population bottleneck, likely due to founder effects associated with the introduction of
*T. clavata* in the US. Subsequent introduction events in the same sample collection range
would appear to have come from the same mitochondrial population of the originally-
introduced *T. clavata*.

Yunnan 1 *T. clavata* represented the only identical non-US COI sequence that grouped
with the USA samples. The closely related Guangdong sample shared sequence similarity

with one nucleotide polymorphism (A/G) at position 527 in the alignment. In addition to the polymorphisms mentioned in *Hoebeke, Huffmaster & Freeman (2015)*, an additional polymorphic site among the COI sequences was observed that distinguished the Yunnan 1/US clade from the other Asian sequences: a C/T polymorphism 339 bp further along the alignment from the previously cited polymorphisms in which the US/Yunnan 1/Guangdong *T. clavata* share a cytosine at that position. Though our results are limited by sample size, the observed mitochondrial haplotype match suggests the founding US population is more closely related to *T. clavata* previously sampled from Yunnan than other samples from Asia used in the current dataset (Fig. 2). It should be noted that the clade representing the *T. clavata* outgroup contains sample Yunnan 2. Yunnan is the most southwest province in China, bordering Myanmar, Laos, and Vietnam, and represents one of the westernmost extents of the native range of *T. clavata*, excluding the Himalayas. The observed phylogenetic discordance of COI within this province, and that observed for COI samples from Zhejiang province, may simply result from limited sample size (only two samples from each region) or an indication of mtDNA diversity within the regions. Regional and population-level analyses of *T. clavata* in its native range are necessary to address questions of genetic diversity and the potential source population for the US founding of *T. clavata*.

The phylogenetic bifurcation observed for the concatenated *Wolbachia* MLST *T. clavata* sequences from the US was unexpected, given the uniform COI sequence identities of the samples (Fig. 4). The single South Carolina sample (CLSC1) grouped with *T. clavata Wolbachia*, which was collected from Guangdong Province (also represented in the COI tree as Guangdong) and was recovered in a clade that included *Drosophila melanogaster*, *Rhagoletis cerasi*, and *Nasonia giraulti* MLST sequences. All Georgia *T. clavata* MLST sequences formed a clade sister to *Agelenopsis aperta* and were recovered in a larger grouping of other spider species, *Mesida yini* and *Araneus ventricosus*, as well as the mosquito *Aedes albopictus*. The apparent phylogenetic discordance of the MLST data from the US suggests *T. clavata* was introduced in the US more than once, with South Carolina and Georgia representing distinct *Wolbachia* populations. The sequence similarities of the Guangdong MLST data to the South Carolina sample and the exact COI sequence match for Yunnan 1 and all US samples may be the result of a source population in southern China, with a relatively uniform mitochondrial background (one COI nucleotide polymorphism) and distinctly different *Wolbachia* populations. Previous empirical and theoretical analyses have established a selective sweep effect by *Wolbachia* on mitochondrial genetic variation, whereby single or repeated waves of *Wolbachia* infection reduce mitochondrial genetic variation (*Turelli, Hoffmann & McKechnie, 1992*; *Kriesner et al., 2013*). *Wolbachia* selective sweeps have also been associated with accelerated rates of molecular evolution for both mitochondria and *Wolbachia* (*Baldo et al., 2010*; *Russell, Saum & Williams, 2022*; *Schulenburg et al., 2000*), potentially complicating conclusions drawn from phylogenetic analyses (*Hurst & Jiggins, 2005*).

Mitochondrial-*Wolbachia* phylogenetic discordance is common among spiders and other arthropods (*Yang et al., 2021*; *Baldo et al., 2008*; *Wendt et al., 2022*; *Russell, Saum & Williams, 2022*). Previous studies of *Wolbachia* infection among spiders in China have

yielded conflicting phylogenetic placement of *T. clavata Wolbachia*, with some studies placing *T. clavata Wolbachia* in the B supergroup (*Wang et al., 2010*) and others in the A supergroup (*Yang et al., 2021*). Incongruent phylogenies among species have been explained as a result of horizontal transfer between species (*Baldo et al., 2008*; *Rowley, Raven & McGraw, 2004*), which appears to be a common inference for incongruent *Wolbachia* phylogenies (*Boyle et al., 1993*; *Heath et al., 1999*; *Vavre et al., 1999*; *Baldo et al., 2008*). Discordant mitochondrial-*Wolbachia* phylogenies, associated with a common mitochondrial genetic background and divergent *Wolbachia* infections, were observed in the spider *Agelenopsis aperta* (*Baldo et al., 2008*). Whether the divergent South Carolina *Wolbachia* strain observed is a subset of a double-infected Georgia strain or is a unique strain itself, horizontal transmission of *Wolbachia* within distinct source populations of *T. clavata* could explain the phylogenetic discordance observed in the present study.

Amplification of the South Carolina sample and analysis of the chromatogram results showed no double peaks characteristic of double infection. In contrast, the Georgia samples all showed double peaks at specific locations corresponding to A and B supergroup reference samples (Fig. 3). We cannot discount the possibility that the MLST alignments used, particularly the Georgia samples, may be compromised by a mix of A and B supergroup sequences. Regardless, the available data suggests a double infection for the Georgia samples and a single infection for the South Carolina sample. Double infection status is also supported by the observation of successful amplification of A- and B-specific wsp sequences. Therefore, without molecular cloning, it was impossible to eliminate the possibility of double-infected *T. clavata* samples from Georgia returning MLST sequences that are not some mosaic of A- and B-supergroup *Wolbachia*. Further, we cannot assume that the South Carolina A-supergroup *Wolbachia* sequence is not identical to an A-supergroup *Wolbachia* sequence in a double-infected Georgia population. However, the status of a single-infected South Carolina population and a double-infected Georgia population appears to distinguish the two populations.

## CONCLUSIONS

The introduction of *T. clavata* to the US, first reported in the state of Georgia (*Hoebeke, Huffmaster & Freeman, 2015*), has resulted in a spreading, established population. Our investigation found the mitochondrial genetic structure of the since-established population in Georgia to be monomorphic for the COI gene region, the same gene region sequence used to first identify the species in the US. Like *T. clavata* in native potential source populations in Asia, the Georgia population was found to be infected with *Wolbachia*. Phylogenetic constructions using COI and *Wolbachia* MLST gene regions found mitochondrial-*Wolbachia* discordance associated with a divergent *Wolbachia* strain from a single sample collected in South Carolina. The presence of evidence for super-infection among Georgia *T. clavata* and a single infection in the South Carolina sample supports the contention that distinct *Wolbachia* populations are present within a uniform mitochondrial background in the area sampled. Future research should expand the sample area within the US and beyond to include the native ranges in Asia. A more comprehensive sampling protocol

would provide valuable population genetic information that could help identify potential source populations for the introduced US population(s) and provide context for the cytoplasmic population structure observed in the introduced population.

## ACKNOWLEDGEMENTS

We appreciate and acknowledge the contribution of Michael Sitvarin and other collectors of *Trichonephila clavata* samples.

### Funding

The authors received no funding for this work. Georgia Gwinnett College STEC 4500 research course funds were provided to James E. Russell and Nicholas R. Mizera. The funders had no role in study design, data collection and analysis, decision to publish, or preparation of the manuscript.

### Grant Disclosures

The following grant information was disclosed by the authors:
Georgia Gwinnett College STEC 4500.

### Competing Interests

The authors declare there are no competing interests.

### Author Contributions

- James E. Russell conceived and designed the experiments, performed the experiments, analyzed the data, prepared figures and/or tables, authored or reviewed drafts of the article, and approved the final draft.
- Nicholas Mizera performed the experiments, analyzed the data, prepared figures and/or tables, authored or reviewed drafts of the article, and approved the final draft.
- Christopher G. Brown analyzed the data, authored or reviewed drafts of the article, and approved the final draft.
- Angela Chuang analyzed the data, authored or reviewed drafts of the article, and approved the final draft.
- David R. Coyle analyzed the data, authored or reviewed drafts of the article, and approved the final draft.
- David R. Nelsen analyzed the data, prepared figures and/or tables, authored or reviewed drafts of the article, and approved the final draft.

### Field Study Permissions

The following information was supplied relating to field study approvals (i.e., approving body and any reference numbers):
Field collection of spiders was approved by Georgia Gwinnett College STEC 4500.

## Data Availability

The COI sequence data is available at GenBank: PV055146–PV055159. The Wolbachia MLST gene sequence data is available at GenBank: PV236029–PV236040 (fbpA); PV236041–PV236052 (hcpA); PV236053–PV236064 (coxA); PV236065–PV236076 (gatB).

## Supplemental Information

Supplemental information for this article can be found online at http://dx.doi.org/10.7717/peerj.19952#supplemental-information.

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
