# Peer review of "Mitochondrial and *Wolbachia* phylogenetics of the introduced Jorō spider, *Trichonephila clavata* (Araneae: Araneidae) in North America"

_PeerJ, doi:10.7717/peerj.19952_

## Round 0.1 · original submission · Minor Revisions

One main issue is phylogenetic construction. For displaying diversity within species, rather than inferring specific phylogenetic conclusions, NJ would be appropriate, but you haven't presented the NJ tree here. Either add it as a Supplementary figure, or remove all reference to it. Your Bayesian analysis does indeed have low clade support, but that is to be expected in a recent time frame. Please emphasise this low clade support more in the Results and Discussion. You might also want to comment that part of the differing patterns of diversity between Wolbachia and the host mtDNA may reflect a higher rate of molecular evolution in the former.

I wonder whether the terms "mtDNA" and "Wolbachia" should appear specifically in the title instead of "cytoplasmic"? When I first read the title, I assumed that the paper was just a (phylogeographic) mtDNA analysis of the spider, since the term cytoplasmic was used in this way in early studies. The Abstract makes it more clear, and these terms should be retrievable from here, but I think it would make the title more informative. Up to you.

Line 104 The term "linkage disequilibrium" is specific to DNA variants in the same genome, usually the same "chromosome". It should not be used to talk about associations of host and parasite genomes. I would use words like "independence" or "associations" instead (if that is indeed what you mean here?).

Lines 57,203,218 Use of possessive (apostrophe s) with Latin names is problematic, since they can be singular or plural. Please remove them and use: "future and history of", "introduction of" "range of" etc instead.

Use direct language in Acknowledgements: "We thank/acknowledge..." etc.

Please check References thoroughly.

Reviewer 1 ·

Basic reporting

Language is concise and easy to understand, concepts are explained clearly, and authors are transparent about the limitations of the study. One suggestion is to briefly explain what double peaks on a chromatogram signify the first time it is mentioned, but this information is explained later, so it is up to the authors' discretion.

Experimental design

Research questions are clear and well-defined. Knowledge gaps to be filled are communicated well. I was curious as to why the authors chose to construct a neighbor-joining tree when they could have used another method, such as maximum likelihood estimation, which is less susceptible to issues such as long-branch attraction. The authors could consider including an explanation for why a neighbor-joining tree was appropriate for their data.

Validity of the findings

Conclusions are clear, and the impact of the study is well stated, with limitations again made transparent.

Additional comments

Some small proofreading comments are as follows:

Line 16: include a comma after "Since its introduction"
Line 19: include a colon after "such as"
Line 64: remove the "a" in "a species"
Lines 107-109: Consider rephrasing these two sentences into one, such as: "Mature female Trichonephila clavata were collected from August to December 2022 from various natural and managed locations before T. clavata invaded the area."
Line 110: "spider" is misspelled as "sprider"
Lines 176-178: consider taking out direct inclusion of accession numbers and instead writing: "(see Table S1 for accession numbers)"
Line 187: remove the commas in "for, at least,"
Line 276: remove the comma after "potential source populations"

Reviewer 2 ·

Basic reporting

-

Experimental design

-

Validity of the findings

-

Additional comments

This study adds a new layer to the understanding of the introduction and spread of the Joro spider (Trichonephila clavata (L. Koch, 1878)) in the SE USA by introducing an analysis of its endosymbiont, Wolbachia. The study reveals that Wolbachia in the Georgia and South Carolina populations are not each other's closest relatives, separating these lineages into two distinct clades. I find this analysis plausible and relevant, less so the COI analysis of the hosts (comment below).

The COI Bayesian tree (Fig. 2) reveals extremely low posterior probabilities. Any values lower than 90% should be deemed as poorly supported. In the presented tree, we see values as low as 18% in the clades of most interest. This must be cautioned as too preliminary to draw conclusions. Slightly better supports, though not very strong ones, are seen in the concatenated Wolbachia tree (Fig. 4).

My other comments target more the phylogenetic and classification issues of Trichonephila. First, the authors follow the spider catalogue that lists nephilines as a subfamily of Araneidae, a vast and undiagnosed assemblage of taxa. Another classification scheme exists that might be more appropriate to be used where Trichonephila and related genera are in the family Nephilidae (https://doi.org/10.1093/sysbio/syy082). Please see recent debates (https://doi.org/10.1093/sysbio/syad021) and either allow for competing classifications or argue for using one. This is not an unimportant issue and may make the paper more durable as it will not get the classification incorrect in the long run. Another error by the authors is the claim that T. clavata and T. clavipes are closely related species. They are congeneric if that means closely related, but are each more closely related to a different set of African species (https://doi.org/10.1093/sysbio/syy082).

In sum, the manuscript, when revised, should represent an important next step in understanding the spread of the Joro spider in the Western hemisphere.

·

Basic reporting

This manuscript meets all of PeerJ's criteria.
I noted minor corrections needed in the references:
In Baldo et al. 2008, Wolbachia should be italicised.
In Serbus et al. 2008, "Annual review of genetics" should be "Annual Review of Genetics"

Experimental design

Why was neighbor-joining used? The tree was not reported or compared to the Bayesian tree. I suggest removing the neighbor-joining analysis as it adds nothing to the manuscript.

How many millions of generations was the Bayesian analysis run for?

Validity of the findings

no comment

Additional comments

In the acknowledgements, I suggest changing "The authors would like to acknowledge..." with "The authors would like to acknowledge..." Also, replace "Trichophila clavata" with "Trichonephila clavata" and italicise it.

---

## Round 0.2 · accepted · Accept

Thank you for your attention to the details of the comments.